# Digitalization, Electricity Consumption and Carbon Emissions—Evidence from Manufacturing Industries in China

**DOI:** 10.3390/ijerph20053938

**Published:** 2023-02-22

**Authors:** Qian Zhang, Qizhen Wang

**Affiliations:** Business School, Nanjing Xiaozhuang University, Nanjing 211171, China

**Keywords:** digitalization, electricity consumption, carbon emissions, manufacturing industries

## Abstract

The development of China’s manufacturing industry is constrained by factors such as energy and resources, and low-carbon development is arduous. Digitalization is an important method to transform and upgrade traditional industries. Based on the panel data of 13 manufacturing industries in China from 2007 to 2019, a regression model and a threshold model were used to empirically test the impact of digitalization and electricity consumption on carbon emissions. The research results were as follows: (1) The digitalization level of China’s manufacturing industry was steadily increasing; (2) The proportion of electricity consumption in China’s manufacturing industries in the total electricity consumption hardly changed from 2007 to 2019, basically maintaining at about 6.8%. The total power consumption increased by about 2.1 times. (3) From 2007 to 2019, the total carbon emissions of China’s manufacturing industry increased, but the carbon emissions of some manufacturing industries decreased. (4) There was an inverted U-shaped relationship between digitalization and carbon emissions, the higher the level of digitalization input, the greater the carbon emissions of the manufacturing industry. However, when digitalization develops to a certain extent, it will also suppress carbon emissions to a certain extent. (5) There was a significant positive correlation between electricity consumption and carbon emissions in the manufacturing industry. (6) There were double energy thresholds for the impact of labor-intensive and technology-intensive manufacturing digitalization on carbon emissions, but only a single economic threshold and scale threshold. There was a single scale threshold for capital-intensive manufacturing, and the value was −0.5352. This research provides possible countermeasures and policy recommendations for digitalization to empower the low-carbon development of China’s manufacturing industry.

## 1. Introduction

Under the goal of carbon neutrality, traditional high-carbon emission industries such as steel and cement have received greater attention, but the digital economy is becoming a new driving force for the high-quality development of China’s economy and plays a very important role in carbon emission reduction. In 2020, the total scale of China’s digital economy reached 39.2 trillion yuan, with a growth rate three times that of China’s GDP and a contribution of 38.6% to GDP. As a new mode of production, digital technology will help China achieve the goals of carbon peaking and carbon neutrality and at the same time, provide fast funding channels for the development of low-carbon cities [1]. Although the development of China’s digital economy is showing a steady upward trend, there is still a phenomenon of regional development imbalance [2].

China is experiencing an unprecedented process of digitalization and modernization, and its manufacturing industry is also speeding up adjustment, optimization, and upgrading. In 2021, the added value of China’s manufacturing industry was 31.4 trillion yuan, accounting for nearly 30% of the world’s total. The energy utilization rate industry continued to rise. The comprehensive energy consumption of steel and other units has dropped by more than 9% compared with 2012. Digitalization and manufacturing are integrated and developed, and the energy efficiency of infrastructure is also continuously optimized. From 2007 to 2019, the level of digitalization in China’s manufacturing industry steadily increased (See Figure 1 for more details).

An important path in achieving carbon emission reduction is transforming the energy industry. In 2020, carbon emissions from national energy consumption accounted for 85% of total carbon emissions, and those from the power sector accounted for 40% of the total. The digital economy mainly empowers the energy sector at three levels to help achieve carbon emission reduction goals. First, from the perspective of the energy supply side, the trend and fluctuation of power demand can be monitored and controlled in real-time through digital technology to achieve the optimal allocation of resources and improve energy utilization efficiency. Second, from the perspective of the energy demand side, digital technology can monitor the disclosure of information such as carbon emissions. It is possible to measure and source carbon emissions, helping companies achieve demand-side management of carbon emissions at a lower cost. This can further improve the carbon emissions trading market. Third, from the perspective of energy trading, digital technology can solve the time and space barriers in the transaction, which can optimize the matching of the supply and demand sides and then improve the energy transactions efficiency.

The rapid development of China’s manufacturing industry is accompanied by a large amount of energy consumption and carbon emissions. Electricity is an important source of energy and is clean, but the process of producing electricity is not. From 2019 to 2021, China accounted for almost all of the growth in global carbon emissions from the power and heat sectors. The CO_2_ emissions from the power and heating sector increased by 6.9% in 2021, due to a sharp increase in global electricity demand. From 2007 to 2019, the proportion of electricity consumption of China’s manufacturing industries in the total electricity consumption hardly changed, maintaining at about 6.8%. However, the total electricity consumption of the manufacturing industry has increased by about 2.1 times, from 122.3 billion kWh to 260.4 billion kWh (See Figure 2 for details). As the world’s largest carbon emitter, China is actively taking responsibility for reducing carbon emissions. “Made in China (2025)” clearly states that by 2025, the added value of carbon emissions per unit of China’s manufacturing industry should be reduced by 40% based on 2015. In recent years, although the carbon emissions of some manufacturing industries have decreased, the total carbon emissions have continued to increase (See Figure 3 for details).

This paper aimed to analyze the impact of digitalization and electricity consumption on carbon emissions in China’s manufacturing industry, accurately identify the influencing factors and threshold effects of carbon emissions in China’s manufacturing industry, and then choose appropriate emission reduction paths and policies to help China’s manufacturing industry successfully realize digitalization and low-carbon transformation goals. Possible marginal contributions are: First, the article analyzes the industry heterogeneity of China’s manufacturing industry and takes into account the spatiotemporal factors of global value chain participation. Second, the article uses the threshold model to analyze the energy, economic, and scale effects of digitalization on China’s manufacturing carbon emissions. Third, the article puts forward specific countermeasures for manufacturing carbon emission reduction from the aspects of demand, supply, and transaction sides.

In the process of digitalization and the rapid growth of energy consumption, China’s manufacturing industry is facing greater pressure in reducing carbon emissions. This paper focuses on the relationship between digitalization, electricity consumption, and carbon emissions. It examines the impact of the digital economy on China’s manufacturing carbon emissions and the threshold effect. Then we propose corresponding carbon emission reduction countermeasures. This paper provides theoretical and practical references for the government and enterprises. Furthermore, it has important practical significance for the development of a low-carbon society.

## 2. Literature Review

### 2.1. Understanding Digitalization

Digitalization involves digital technology and its integrated use in the production process [3]. The development of the digital economy includes two aspects. One is digital industrialization. Information technology has given birth to many new industries. The industry based on digital elements has promoted the industrial structure to be technology-intensive and environment-friendly. The second is industrial digitalization, which refers to the combination of traditional industries and digitalization, and the application of digitalization in production to promote the transformation and upgrading of traditional industries. The upgrading of industrial structures can promote the use of clean energy, replace traditional high-carbon emission energy with clean energy, and ultimately reduce carbon emissions [4]. The upgrading of industrial structures can stimulate the R&D and application of low-carbon technologies. Meanwhile, it can improve the energy structure to play a better substitution role and promote the green transformation of enterprises [5].

Digitalization has a greater impact on carbon emission efficiency. It can improve carbon productivity, and its impact on the central and western regions of China is significantly greater than that on the eastern regions. Furthermore, it mainly affects carbon productivity through technological innovation and industrial structure optimization and upgrading [6]. The promotion effect of digitalization on carbon emission reduction shows a trend of increasing with time and has a positive spatial spillover effect. Digitalization is becoming one of the new sources of energy to improve green development. With the technology accumulation, the coefficient of the impact of digitalization on total factor carbon productivity is getting higher and more significant [7].

However, the carbon-reducing effect of digitalization is controversial. Some scholars [8,9,10] believe that digitalization has a carbon emission reduction effect, while others [11,12] believe digitalization can promote carbon emissions. Therefore, we will discuss these questions in the next section. The topics are divided into the decoupling of digitalization and carbon emissions, the uncertainty of digitalization and carbon emissions, the relationship between digitalization and manufacturing carbon emissions, the specific path to realize digital carbon emission reduction, and finally puts forward the *hypotheses* that this paper wants to test.

### 2.2. Debate on the Relationship between Digitalization and Carbon Emissions

#### 2.2.1. Digitalization and Carbon Emissions Are Gradually Decoupling

(1)Linear analysis

Based on the linear analysis between digitalization and carbon emissions, it is found that the two are slowly decoupling. Digitalization mainly promotes the low-carbon transformation of cities through innovation, and its impact on low-carbon development in cities will become stronger and stronger [13]. Digitalization uses the internet to reduce offline activities, travel, and carbon emissions. Meanwhile, it promotes the popularization of green and low-carbon behaviors, making low-carbon a daily behavior standard and also promoting the effective use of a green economy [4]. From a technical point of view, China’s information and communications technology (ICT) industry helps reduce carbon emissions, and the ICT industry in the central region has a greater impact on CO_2_ emissions than the eastern region [14]. The innovative development of ICT provides opportunities for the coordinated development of shared prosperity, energy conservation, and emission reduction. It effectively promotes carbon emission reduction by reducing energy consumption [15]. Meanwhile, improving energy structure and technological progress can effectively reduce carbon emission intensity [4]. Digitalization can significantly increase carbon productivity. Technological innovation, reduction of energy consumption intensity, and improvement of urban productivity are the main paths [16,17].

Digitalization has a significant driving effect on the coordinated governance of carbon dioxide and haze pollution, and there is a positive spatial spillover effect [18]. It mainly improves environmental pollution through technological innovation and optimal allocation of resources [19]. Furthermore, there is a long-term positive and significant relationship between internet use and carbon emissions, but no causal relationship exists. The rapid growth of the internet is not the main reason the environment is threatened. Therefore, promoting the development of the internet will not lead to environmental degradation [20].

(2)Nonlinear analysis

In order to more scientifically assess the relationship between the digital economy and carbon emissions, nonlinear analysis is becoming more popular. In the study of digitalization and carbon emissions, the development of regional digitalization has significantly reduced the intensity of carbon emissions. The relationship with carbon emissions presents an inverted U-shaped relationship that first rises and then declines. The specific transmission paths are mainly technological innovation, industrial structure, and energy structure. Digitalization has latecomer advantages in achieving carbon neutrality goals [21]. The empirical analysis finds that comprehensive infrastructure construction will increase energy intensity and thus hinder carbon emissions. However, information integration infrastructure is conducive to developing the tertiary industry, and the carbon emissions generated will be less than those generated by comprehensive infrastructure construction. This leads to an inverted U-shaped relationship between integrated infrastructure development and carbon emissions [22]. However, digitalization contributes to carbon emissions when green energy is less efficient and vice versa [23].

Besides, digitalization has spatial spillover effects on carbon emission reduction. Using the spatial Durbin model (SDM), it is found that digitization has a U-shaped spatial spillover emission reduction effect and presents an inverted U-shaped carbon emission reduction effect that is first promoted and then suppressed. Technological progress and economic growth are the main mechanisms [24]. Using the panel data of 277 cities in China from 2011 to 2019, an inverted U-shaped nonlinear relationship between digitization and carbon emissions was also found. The industrial structure upgrading makes the effect of digitalization on carbon emissions also follow the characteristics of the Environmental Kuznets Curve [25]. Digitalization has a significant negative direct effect on green total factor energy efficiency (GTFEE) through electrification, hollowing out of industrial scale, and industrial efficiency. However, with economic development, its impact on GTFEE gradually turns from negative to positive. Based on the SDM and threshold models, the inverted U-shaped relationship between digitalization and carbon emissions has been further verified [26].

As a new form of economy, digitalization is important for reducing carbon emissions in the transportation and logistics industries. It has a mitigating effect on carbon emissions in the transportation sector. It also accelerates carbon emissions in the transportation sector in the low-urbanization stage but reduces carbon emissions in the high-urbanization stage [27]. With the provincial panel data from 2005 to 2019, the nonlinear regression model and the quantile regression model were used to empirically test the U-shaped relationship between digitalization and carbon emissions in the logistics industry. In the first half of the U-shaped relationship, digitalization had both a restraining effect and a significant evolutionary effect on carbon emissions in the logistics industry. As the quantile increases, the marginal impact of digitalization on carbon emission reduction in the logistics industry gradually decreased [28].

#### 2.2.2. Digitalization Brings Uncertainty about Carbon Emissions

Although the above studies find that digitalization and carbon emissions are decoupling, the nexus between the two remains uncertain. In analyzing the carbon deduction effect of digitalization, considering the impact of digital demand and supply, digitalization may bring about 6% of carbon emissions [11]. With the promotion of digital demand and scale, between 2002 and 2007, the carbon emissions brought by digitalization rose from 210 Mt to 418 Mt. From 2007 to 2017, with the improvement of carbon efficiency and digital application structure, the carbon emission caused by digitalization has been alleviated. However, with the intermediary model and the panel threshold model, it was found that improving energy efficiency can promote carbon emission reduction, although digitalization increases carbon emissions. Nevertheless, digital development is not conducive to the improvement of energy efficiency. Considering energy efficiency, digital development has a significant double-threshold effect on carbon emissions, showing an N-shaped trend. Population expansion, coal-based energy consumption structure, and industrial structure were the main reasons for the increased carbon emissions [29].

Although digitalization can effectively reduce urban carbon emissions and improve total factor productivity, the improvement of energy efficiency, technological innovation, and industrial structure upgrading are the main reasons for the existence of digital low-carbon governance effects. However, digitalization can only promote the low-carbon transformation of old industrial bases. The urban development of traditional resource industries is path-dependent, and the effect of low-carbon governance is not obvious [30]. Although artificial intelligence can produce carbon emission reduction effects through industrial structure, information infrastructure, and green technology innovation, these are only for big cities and cities with better infrastructure and advanced technology. There are differences in the development of the digital economy among different countries, especially in hyper-digitalized and under-connected countries. Although digitalization reduces total carbon emissions, it increases carbon emissions per capita [31].

### 2.3. Digitalization, Electricity Consumption and Manufacturing Carbon Emissions

#### 2.3.1. Digitalization and Manufacturing Carbon Emissions

Digitalization is a very important industrialization process. Influenced by Industry 4.0, it directly affects all production processes and manufacturing sectors. Therefore, it is imperative to increase the productivity and sustainability of the manufacturing sector [32]. Manufacturing digitalization is an important enabling factor for improving competitive advantage [33]. Industry 4.0 has become a continuous and predicted outcome of past industrial ages. From a technological point of view, it can be considered an increase in digitalization and automation, as well as an increase in communication enabled by the creation of digital value chains [34]. However, digital traceability should be used to improve the traceability and added value of products, shorten the production cycle and promote the manufacturing industry to adapt to the requirements of Industry 4.0 [35]. 

The reduction of carbon emissions is the main reason driving the growth of TFP and technical efficiency in the context of the deep integration of the digital service industry and the manufacturing industry. Industrial integration and carbon emissions show a U-shaped relationship. The integration of capital-intensive, technology-intensive, and labor-intensive manufacturing industries and digital services promotes the growth of total factor productivity but first suppresses and then promotes carbon emissions [36]. Digitalization has a positive effect on TFP in time and space and can promote the development of manufacturing. Labor-intensive and capital-intensive industries have the same characteristics as the total sample [37].

The development of digitalization reduces the carbon emission intensity of enterprises and improves the efficiency of resource allocation. However, the market drive has improved the ability of digital carbon emission reduction, while government regulation has reduced the ability of digital carbon emission reduction [38]. Through the sample of heavily polluting enterprises listed on China’s A-shares, it is found that digitalization can significantly improve the efficiency of energy conservation and emission reduction of enterprises, especially for mining and manufacturing industries. Digitalization has promoted cleaner production through technological innovation, easing financing constraints, and promoting market competition. This promotion effect is more significant in areas with more developed economies and less government financial pressure [12]. From a global perspective, investment in manufacturing digitization has reduced carbon emission intensity. The industry spillover effect becomes more significant over time. From the perspective of industry heterogeneity, the carbon emission reduction effect of digitalization in pollution-intensive manufacturing is more obvious [39].

Based on this, this paper proposes the following hypothesis:

**Hypothesis** **1.***There is an inverted U-shaped relationship between digitalization and China’s manufacturing carbon emissions*.

#### 2.3.2. Electricity Consumption and Carbon Emissions

Increased electricity use adversely impacts carbon emissions [40,41,42,43,44,45]. The continuing increase in electricity consumption is one of the main sources of carbon emissions [40]. Electricity-related carbon emissions release more than 40% of global and Chinese carbon emissions [44]. Electricity is usually an important energy source for a country, and the demand for electricity often increases energy consumption and pollution. Through the econometric analysis of the co-integration panel, it is proved that the adverse effect of electricity consumption on carbon emissions exists. However, electricity output from renewable sources can ease the pressure on carbon emissions. Overall, electricity consumption and generation are the main sources of carbon emissions [41].

China is also committed to research on carbon reduction of electricity consumption. Regression analysis of 25 years of relevant data from 123 countries found that using renewable energy had a carbon emission reduction effect [42]; however, there was no causal relationship between electricity consumption and carbon emissions in China [46]. In 2020, 66% of China’s power generation came from coal, and coal consumption accounted for 61% of energy consumption. A reduction in coal consumption would result in a 51% reduction in overall carbon emissions [43]. Based on the carbon release effect of electricity consumption, how to reduce carbon emissions through technology has attracted the attention of the power sector. Previous studies have often neglected electricity-related carbon emissions induced by electricity consumption. Through the comprehensive application of IPCC’s carbon emission accounting method, taking Shanghai as the research object, it was again verified that electricity consumption and population size have indeed promoted carbon emissions. However, the increase in electricity efficiency and the decline in carbon emission intensity offset the increase in carbon emissions [45]. As the digital transformation deepens, the promotion effect of energy consumption per capita on carbon emissions is weaker, while the effect of renewable energy on carbon emission reduction is stronger [47].

**Hypothesis** **2.***Electricity consumption contributes to China’s manufacturing carbon emissions*.

### 2.4. The Specific Path of Digital Empowerment for Carbon Emissions Reduction

Digitalization empowers carbon emissions mainly through the following paths. First, it mainly relies on industrial progress and energy consumption optimization to curb carbon emissions and has significant spatial spillover effects on neighboring provinces [48]. In the short term, increased energy consumption and non-green technological progress are the main paths to increasing carbon emissions. However, technological progress and industrial structure upgrading are the main paths for long-term carbon emission reduction [49]. Second, it plays a role mainly through resource flow and energy flow. From the perspective of element resource misallocation (capital misallocation and labor misallocation), digitalization can improve carbon emission efficiency in both southern and northern China. Meanwhile, it has a long-term positive impact on the carbon emission rate by mitigating factor misallocation [50]. Digital financial inclusion is an important factor for digitalization to affect carbon emissions [51]. Third, from the perspective of digital transformation, digital infrastructure, digital trade competitiveness, digital technology, and energy consumption have significant threshold effects on carbon emissions. Although trade brings economic benefits, it also implies the environmental costs of carbon emissions [44]. Smart city construction will also significantly reduce corporate carbon emission intensity [52]. Finally, improved energy efficiency helps reduce carbon emissions in manufacturing and transportation [53]. 

The above analysis found that there are many related studies. However, the relationship between digitalization and carbon emissions is still inconclusive. Furthermore, although electricity consumption boosts carbon emissions, there are not many specific analyses on the manufacturing industry. Based on this, this paper builds the analysis framework shown in Figure 4 to capture the direct effects of digitalization on carbon emissions and related threshold effects (See Figure 4 for more details).

## 3. Methodology and Data Source

### 3.1. Variables and Data Sources

According to the results of our literature review, we found that the carbon emissions of the manufacturing industry are mainly affected by factors such as energy consumption, scale effects, industrial structure transformation and upgrading, international trade and division of labor under an open economy. At the same time, factor input plays a very important role in exerting the competitive advantage of digital transformation in the manufacturing industry. Factor input can promote product upgrading, reduce undesired output, and reduce pollution. Based on the theoretical mechanism, this paper takes carbon emissions as the explained variable and digital input level and electricity consumption as the core explanatory variables. The latter is used to measure the energy effect. Taking into account other different effects and data availability, control variables, including industry factors, industry scale, export dependence, import dependence, the industry added value, participation in global value chains, the proportion of industrial waste gas treatment cost, are selected. The variable GVC-pat was used to measure the upgrading of industrial structure because participation in GVC will promote the change of industrial structure [54]. Besides, China is facing pressure to reduce carbon emissions, but at the same time it is facing the heavy responsibility of economic development. At the time of carbon emission reduction, we must consider the cost of pollution control. Table 1 presents all selected variables and data sources.

### 3.2. China’s Manufacturing Sector Segmentation

The manufacturing industries selected in this article were the 13 manufacturing sub-sectors covered by the ADB MRIO2021. The sub-sectors with missing data were deleted. To connect with the manufacturing industry in China Statistical Yearbook and China Industrial Statistical Yearbook, this paper processed relevant data, and the processing methods are shown in Table 2. 

The processing method for the industry: 1. Merge the rubber and plastics industries into rubber and plastics; 2. Consolidate automobile manufacturing and railway, ship, aerospace, and other transportation equipment into transportation equipment manufacturing; 3. Consolidate the food processing industry, food manufacturing, beverage manufacturing, and tobacco processing industries into food, beverage, and tobacco; 4. Merge the paper and paper products industry and the printing industry, the reproduction of recording media into pulp, paper, paper products, printing, and publishing; 5. The ferrous metal smelting and rolling processing industry, non-ferrous metal smelting and rolling processing industry, and metal products industry were combined into basic metals and fabricated metal.

### 3.3. Model Construction

#### 3.3.1. Panel Regression Model

In order to verify the impact of digital investment and electricity consumption on carbon emissions, this paper selected the panel data of 13 industries in China’s manufacturing industry from 2007 to 2019, and finally built the following panel data model:(1)$$\;\;\;Carbon_{it}=\alpha +\beta Dig_{it}+\beta Dig_{it}^{2}+\rho Er_{it}+\gamma Cont_{it}+\mu_{i}+\delta_{t}+\varepsilon_{it}$$
where $Carbon_{it}$ represents the total carbon emission of manufacturing industry *i* in period t, and α represents the constant item. Dig represents the core explanatory variable digital input level. To verify the nonlinear impact of digitalization on carbon emissions, the variable *Dig^2^* was included to represent the square of the digital input level; Er represents the proportion of electricity consumption of each industry in the total manufacturing electricity consumption; *Cont* represents the control variable; $\mu_{i}$ and $\delta_{t}$ represent the individual and period effects respectively; and $\varepsilon_{it}\;$ represents the stochastic disturbance term.

#### 3.3.2. Threshold Effect Model

The effect of digitalization on carbon emission reduction mainly depends on a sound digital ecology and industrial layout. Bridging the existing digital division is an important problem to be solved in the process of digitalization. Digital technology empowers carbon emission reduction mainly through energy structure, economic base, and industrial scale. Therefore, this paper used the threshold effect model to further explore the energy, economic, and scale effects of digitalization on China’s manufacturing carbon emissions. The specific threshold model was constructed as follows:(2)$$Carbon_{it}=\alpha +\beta Dig_{it}\cdot I(thr_{it}<\gamma )+\delta Dig_{it}\cdot I(\gamma \leq thr_{it})+\theta Cont_{it}+\mu_{i}+\varepsilon_{it}$$
where $Carbon_{it}$ represents the explained variable. *Dig* represents the level of digitalization, and *thr* represents the threshold variables, which are the proportion of electricity consumption, the economic added value of the manufacturing industry, and the industry scale; *I* (·) represents the indicator function; γ represents the threshold value to be estimated; *Cont* represents each control variable, and the control variables are consistent with the benchmark model; $\mu_{i}$ represents the industry effect; and $\varepsilon_{it}$ represents the stochastic disturbance term.

## 4. Results

### 4.1. Descriptive Statistics of Variables

To avoid the influence of different dimensions of variables, all the variables were standardized, and Table 3 displays their descriptive statistics. In this paper, variance inflation factor (VIF) and tolerance (1/VIF) were used to test whether there was multicollinearity among the variables. The higher the VIF value, the more serious the multicollinearity. The tolerance was generally between 0 and 1. The smaller the tolerance, the more serious the collinearity. The average value of the selected variable VIF in this paper was 8.99, and the tolerance was 0.22, indicating that there was no multicollinearity among the variables.

### 4.2. Estimations Results

Column (1) of Table 4 shows the model estimation results. There is an inverted U-shaped relationship between digitalization and carbon emissions, the higher the level of digitalization, the greater the carbon emissions of the manufacturing industry. Nevertheless, when digitization develops to a certain extent, it will curb carbon emissions. However, electricity consumption and carbon emissions in the manufacturing industry show an increasing trend with a positive correlation. 

The higher the electricity consumption, the more carbon is emitted. This is because electricity consumption is a demand for energy. High power usage leads to high power consumption and more energy consumption [57]. Although electricity is a secondhand energy source, it can be regarded as a clean energy source. However, the generation process of electricity is accompanied by primary energy use such as oil, coal, natural gas, etc., which will bring pollution; however, the process of generating electricity from water, nuclear energy, and wind energy is clean. China’s economic development needs energy, and its electricity demand will grow by 10% in 2021. Coal is required to meet the 56% increase in electricity demand as this growth exceeds the growth of low-emissions supply. Although China is vigorously developing new energy sources, new energy sources are far from meeting today’s market demand, and coal investment is still the main force for power generation. The demand for electricity consumption directly or indirectly promotes electricity production, thereby causing carbon emissions in various industries [58]. China’s non-clean energy power generation accounts for about 65% in 2021. Because of the energy structure and energy efficiency, the power sector still has a certain distance from zero emissions [59]. The reasons may be: first, people’s living standards are increasing, and the demand for electricity is increasing, resulting in more carbon emissions from electricity; second, the loss in the power transmission process is large, and energy waste is serious, which also brings about an increase in carbon emissions.

Manufacturing industry factors have a negative correlation with carbon emissions, which is significant at the 5% level. The higher the ratio of paid-in capital to labor factors in the manufacturing industry, the lower the total carbon emissions. The production strategy of a capital-constrained manufacturing producer would be less sensitive to carbon prices and would re-engineer older products at higher quality levels. However, manufacturers with high industry factors and abundant funds will be more sensitive to carbon prices and carbon emissions when formulating production strategies. Therefore, there will be a more obvious carbon emission reduction effect [60].

The manufacturing industry scale is negatively correlated with carbon emissions and is significant at the 1% level. It shows that China’s manufacturing industry has the potential to reduce carbon emissions. While the scale of the industry is expanding, the industrial structure is also constantly being adjusted. Additionally, the proportion of high-emission enterprises will become lower and lower. The scale of the manufacturing industry and carbon emissions will eventually be decoupled. Large-scale production under economic agglomeration is conducive to improving the emission reduction effect of diversification [61].

The increase in export dependence can promote carbon emission reduction and is significant at the 1% level. The carbon emission factors brought about by import and export trade are becoming more significant. The reduction in exports is mainly due to consumption-based carbon emissions [62,63]. As China’s exports have shifted from primary industrial products to high-tech products, the proportion is increasing. The technical effect of export products is becoming more and more obvious. Compared with the extensive production method, exports are conducive to technological innovation and the exertion of carbon emission reduction effects.

The increase in import dependence promotes carbon emissions and is significant at the 1% level. The diversification of imported and exported products will affect energy-related consumption. One of the main goals of manufacturing development in developing countries is to increase energy efficiency. Imports can introduce new and improved technologies, update production methods, and further promote green production. However, the diversification of imported products also makes it easier to obtain cheaper intermediate products. If these intermediate products include building materials, mechanical appliances, and transportation equipment, they will increase the total carbon emissions of developing countries [64]. Besides, to meet domestic demand by importing more energy-intensive products will bring carbon emissions into the importing country and increase the carbon emissions of the importing country. In the process of reducing carbon emissions, special attention should be paid to the hidden carbon emissions in trade and the resulting environmental game between governments [61].

The output value of the manufacturing industry is positively correlated with carbon emissions, but insignificant. It shows that the decoupling of carbon emissions and economic development has not been verified. China’s economic development model has changed from a planned traditional economy to a socialist market economy. Under the tide of de-industrialization, China has gradually increased its efforts in the reform and opening-up of the economy. In the context of expanding domestic demand, infrastructure construction has increased, and large-scale manufacturing clusters have gradually formed in the coastal areas. China has gradually become the world’s largest exporter. Under the goal of manufacturing transformation and upgrading, the relationship between industry-added value and carbon emissions has gradually improved. However, increasing infrastructure construction gradually increases carbon emissions [65].

The increase in the global value chain (GVC) participation can promote carbon emission reduction and is significant at the 1% level. With the continuous improvement of global integration, participation in the GVC has become the new normal of the international labor division. The low status of China’s manufacturing participation will lead to the high carbonization of enterprises’ production methods. The rise of GVC participation greatly impacts the manufacturing industry‘s carbon emissions. The increase in intermediate product exports can promote the rise of GVC status, which is more conducive to reducing carbon emissions [66,67].

The cost of industrial waste gas treatment is positively correlated with carbon emissions and is significant at the 1% level. China is faced with the dual tasks of reducing carbon emissions and developing the economy. On the one hand, to obtain environmental governance performance, there may be a development path of pollution first and then governance. On the other hand, pollution control equipment may have high energy consumption. In addition, considering the spatial spillover effects of environmental governance, pollution governance does increase local carbon emissions [68].

### 4.3. Robustness Test and Heterogeneity Analysis

#### 4.3.1. Robustness Test: Core Explanatory Variables Replacement

This paper adopted the method of [69] to measure the digital input level. The digital input level was measured by the complete consumption coefficient; that is, the direct consumption coefficient plus the indirect consumption coefficient. This article started with the industries that the digital economy relies on. This includes the investment in software services and digital economic hardware facilities. Then, based on the input-output model, the complete consumption coefficient of each manufacturing industry was calculated; that is, the total input amount of the industries relying on the digital economy that needs to be consumed to produce the final products. This paper evaluated the digitalization level of the industry by calculating the complete consumption coefficient of the digital industry in various industries in the manufacturing industry.

The calculation method of the direct consumption coefficient was as follows. Variable *a_ij_ (i,j = 1,2,3,…, n)* indicates the value of the goods or services of the department *i* directly consumed by the unit total output of department *j* in the production process. It is usually expressed as matrix A. The complete consumption coefficient is the sum of direct consumption and indirect consumption, usually expressed by B. The larger the direct coefficient, the stronger the direct dependence of department *j* on department *i*.
(3)$$a_{ij}=x_{ij}/X_{j}\;\;(i,j=1,2,3,\ldots ,n)$$
(4)$$b_{ij}=a_{ij}+\sum_{k=1}^{n}a_{ik}a_{kj\;}+\sum_{s=1}^{n}\sum_{k=1}^{n}a_{is}a_{sk}a_{kj}+\ldots$$
where in the first item, $a_{ij}$ represents the direct consumption of the *j* product department to the *i* product department. The second item $\sum_{k=1}^{n}a_{ik}a_{kj\;}\;$ indicates the first round of indirect consumption of product department *j* to product department *i*; $\sum_{s=1}^{n}\sum_{k=1}^{n}a_{is}a_{sk}a_{kj}\;$ is the second round of indirect consumption, and so on. Round *n + 1* is the indirect consumption of round *n* (Please see Formulas (3) and (4)). Therefore, the full consumption coefficient was used to measure the digital input level of each manufacturing industry, which was brought into the benchmark model. The model estimates are in Column (2) in Table 4, which proves the model estimation results were robust.

#### 4.3.2. Heterogeneity Analysis

In this paper, the manufacturing industry was further divided into capital-intensive, labor-intensive, and technology-intensive. Afterward, two digital evaluation indicators were used for model estimation. Table 5 shows the estimated results. The empirical results show that although the relationship between capital-intensive digitization and carbon emissions shows a U-shaped nonlinear relationship, it was not significant. The higher the level of capital-intensive digitalization, the lower the carbon emissions will be. There was a U-shaped nonlinear relationship between labor-intensive, technology-intensive, and carbon emissions. However, for all manufacturing industries, the higher the electricity consumption, the greater the carbon emissions, and this result was significant at the 1% level.

### 4.4. Endogeneity

In this paper, the method of IV-2SLS was used to alleviate the possible endogeneity problems in variables. In this paper, the first-order lag item of the core explanatory variable (Dig) was used as an instrumental variable, and it was brought into the benchmark model for verification. The results prove that the model was robust. There was a positive linear relationship between electricity consumption and carbon emissions (See Table 6 for details).

### 4.5. Threshold Effect

Table 7 shows the estimated results of the threshold test and the threshold value. The results show that the digitalization of labor−intensive industries had double thresholds for energy effects on carbon emissions, and the thresholds were −0.5478 and 0.0042, respectively. Moreover, there was a single economic threshold and industry scale effect threshold for the impact of labor-intensive industry digitization on manufacturing carbon emissions, and the threshold values were −0.4233 and −0.8088, respectively. Capital-intensive industries had a single scale threshold, and the threshold value was −0.5352; Technology-intensive industries had double energy thresholds, and the threshold values were −0.2248 and 0.8068 respectively. In addition, there was a single economic threshold and industry scale threshold simultaneously, and the threshold values were −0.2786 and 0.1690, respectively. 

Table 8 shows the results of estimating the three groups with thresholds. For labor-intensive industries, when power consumption reaches the threshold, digitalization can promote China’s manufacturing industry’s carbon emissions. However, when the power consumption exceeds the threshold, although the coefficient of digital investment is still positive, the coefficient becomes smaller. This means that electricity consumption has less impact on contributing to carbon emissions as technology improves. The same effect exists for economic value added; when the economic value added exceeds the threshold, the impact of digital investment on carbon emissions will gradually become smaller. In terms of the industry scale, before and after reaching the threshold, the expansion of the industry scale has a restraining effect on the carbon emissions of labor-intensive industries, and the restraining effect is gradually improved. For capital-intensive industries, the expansion of the industry scale has a positive effect on the carbon emissions of capital-intensive industries, but as the industry scale exceeds a certain threshold, this promotion effect is gradually alleviated. For technology-intensive industries, before and after electricity consumption reaches a threshold, digital investment can curb China’s manufacturing carbon emissions. In contrast, as electricity consumption exceeds a certain threshold, this inhibitory effect will gradually slow down. Economic value-added can promote the increase of carbon emissions, but as the economic value-added exceeds the threshold, the promotion effect of digitalization on carbon emissions is greatly alleviated. The scale of the industry also promotes the increase of carbon emissions, but as the scale of the industry exceeds a certain threshold, the impact on carbon emissions gradually weakens.

## 5. Discussion and Conclusions

### 5.1. Discussion

Existing studies have analyzed the correlation between digitalization and carbon emissions. For example, Yang et al. [21] found that digitalization can reduce carbon emission intensity, and this effect has obvious regional heterogeneity. Wang et al. [70] found that digital technology is an important path to achieving carbon neutrality. Moreover, digitalization mainly reduces carbon emissions by optimizing resource allocation and reducing energy consumption costs. Although the existing research is sufficient, the analysis of digital investment and electricity consumption in the manufacturing industry has been little, thereby buttressing the innovation of our research. Besides, this research takes into account the factors that the manufacturing industry participates in international competition, that is, the degree of participation in the global value chain. Only by participating in international competition can we force the transformation and upgrading of industrial structure, which is ultimately conducive to carbon emission reduction [54].

The estimation results of this research are consistent with results in [21,57]. The higher the level of digitization, the greater the carbon emissions of manufacturing. The demand for electricity contributes to carbon emissions. The main reasons are as follows. First, in the short term, the increase in energy use and non-green technologies use have caused an increase in carbon emissions. In the long run, the upgrading of the industrial structure and the spillover of technology have made digitization show a strong carbon emission reduction effect [71]. Second, the increase in demand for electricity consumption has brought about an increase in electricity supply and coal energy consumption, making electricity consumption a major contributor to carbon emissions [40,43].

### 5.2. Conclusions

This paper mainly used the panel regression model to analyze the impact of digital input and electricity consumption on China’s manufacturing industry’s carbon emissions and industry heterogeneity. It used the threshold model to analyze the energy, economic, and scale effects of the manufacturing industry. The conclusions were as follows:

(1) The level of digital investment in China’s manufacturing industry is rising steadily. From 2007 to 2019, the proportion of electricity consumption of China’s manufacturing industries in the total electricity consumption hardly changed, maintaining at about 6.8%. However, the total electricity consumption of the manufacturing industry increased by about 2.1 times. From 2007 to 2019, the total carbon emissions of China’s manufacturing industry increased, but the carbon emissions of some manufacturing industries decreased.

(2) Hypothesis 1 was verified. There was an inverted U-shaped relationship between digitization and carbon emissions. The higher the level of digitization, the greater the carbon emissions of manufacturing; however, when digitization develops to a certain extent, it will curb carbon emissions.

(3) Hypothesis 2 was verified. There was a significant positive correlation between electricity consumption and carbon emissions in the manufacturing industry. The higher the electricity consumption, the more energy is consumed and the more carbon emissions are generated.

(4) The impact of digitalization of labor-intensive and technology-intensive manufacturing on carbon emissions da double energy threshold, a single economic threshold, and an industry scale threshold. There was a single scale threshold for capital-intensive manufacturing, and the threshold value was −0.5352.

According to these research results, this paper puts forward the following policy implications.

(1) From the perspective of the supply side, energy structure transformation and industrial optimization and upgrading should be promoted to improve energy utilization efficiency. Optimizing the energy structure is an important way for the low-carbon development of the manufacturing industry [65]. Carbon emissions have a great impact on developing a green economy. Carbon dioxide emissions do reduce environmental performance and green economic performance. However, digital development, technological innovation, and industrial structure upgrades can promote green economic performance [72]. China is the largest energy consumer and carbon emitter in the world. Optimizing industrial structure, reducing population size, and adjusting energy structure can indeed promote carbon emission reduction [73].

(2) From the perspective of the demand side, the total energy consumption should be reasonably controlled to improve the efficiency of energy-intensive utilization through digitization. Digital technology is an effective path to achieving carbon neutrality, and it mainly reduces carbon emissions by optimizing resource allocation and reducing energy consumption costs [70]. The development of the manufacturing industry has increased the importance of digital elements and processes in strategy and planning. The concepts of digitization and automation are distinct yet interrelated. Both can be used directly in the manufacturing field. Digitization can generate large amounts of data and form network integration. Automation can improve inefficient production steps and increase the consistency of the production process. The development of China’s manufacturing industry should improve the ability of independent innovation and take the road of innovation and development [74]. In terms of dependence on foreign trade, the self-sufficiency rate of China’s high-end chemical products is insufficient, and government policy support is needed to increase production capacity and reduce dependence on foreign trade.

(3) From the transaction point of view, the market mechanism is the most direct path to achieve the optimal allocation of resources and the most cost-effective way to achieve carbon emission reduction. Digitalization’s suppression of carbon emissions has a phased feature. Under the blockchain technology of big data, the establishment of a carbon emission trading rights platform should be improved [75]. Carbon emission trading and carbon tax mechanism are the main ways to achieve carbon emission reduction. Excessive carbon pricing will reduce the economic advantages of carbon transaction costs for high-emission manufacturing enterprises [76]. Digital technology has played an important role in reducing carbon emissions from regional trade and improving the energy efficiency of traded products. Trade has the potential to affect embodied carbon emission flows and embodied carbon emission intensity. The share of carbon emissions from trade is declining, and the manufacture of computers and electro-optical products is the main source of embodied carbon emissions [77]. In China, the ICT sector can generate a large volume of emissions through the demand for carbon-intensive intermediate inputs in the non-ICT sector. Moreover, the power and basic materials sectors are significant sources of carbon emissions; therefore, addressing ICT-related carbon emissions requires a targeted, integrated carbon management strategy that combines supply chain and economic drivers [78]. Consequently, the realization of low-carbon development in the manufacturing industry requires the joint participation of the government, market, enterprises, and technical service departments.

The limitations of this article are: First, due to the availability of data, this article only analyzes the situation of 13 manufacturing industries in China from 2007 to 2019. Second, there is no further in-depth analysis of the energy efficiency and energy consumption structure of the manufacturing industry. This is also our future research direction.

## Figures and Tables

**Figure 1 ijerph-20-03938-f001:**
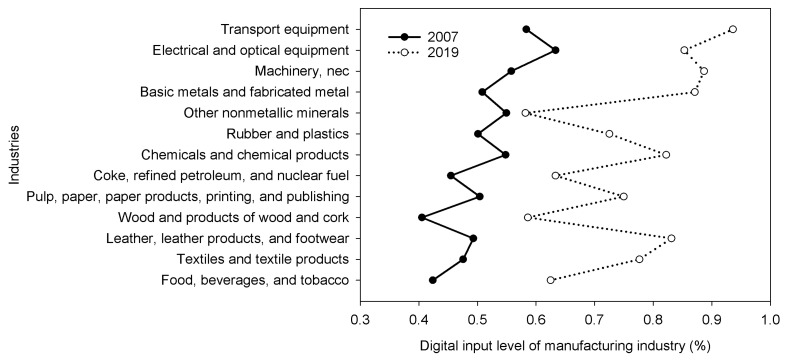
Digitalization level of China’s 13 manufacturing industries in 2007 and 2019 (%).

**Figure 2 ijerph-20-03938-f002:**
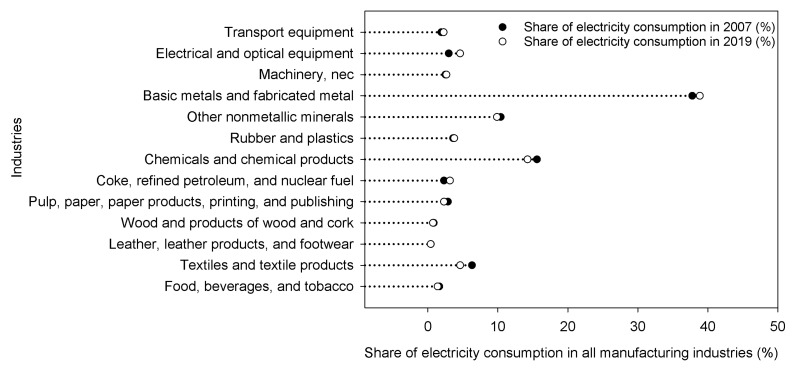
The proportion of electricity consumption in China’s 13 manufacturing industries in 2007 and 2019.

**Figure 3 ijerph-20-03938-f003:**
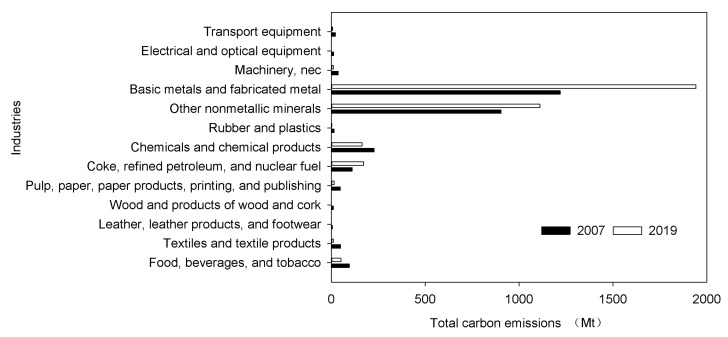
Total carbon emissions of China’s 13 manufacturing industries in 2007 and 2019.

**Figure 4 ijerph-20-03938-f004:**
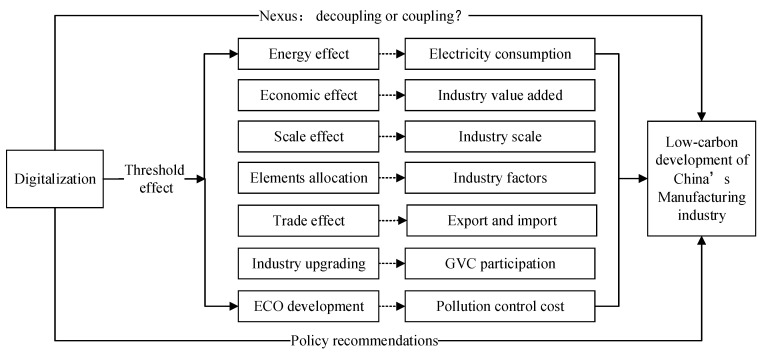
Research framework and theoretical mechanism.

**Table 1 ijerph-20-03938-t001:** Variables and data sources.

	Abbreviation	Variable	Measurement	Data Source
Dependent variable	Carbon	Total carbon emissions	Data aggregation	CEADs
Core explanatory variable	Dig	Digital level	Value added of C14 (electrical and optical equipment) and C27 (post and telecommunications)/Gross Value Added in the Manufacturing Sector	ADB MRIO2021
	Dig^2^	Square of Dig	Dig×Dig	ADB MRIO2021
	Er	Proportion of electricity consumption in the industry	Electricity consumption of each manufacturing industry/total electricity consumption of manufacturing industries	IFIND
Control variable	Ind	Industry factors	Ratio of capital factor to labor factor (Manufacturing industry paid-in capital divided by manufacturing employees)	CISY, CSTSY
	Scale	Industry scale	The output value of each manufacturing industry divided by the total output value	ADB MRIO2021
	Exp	Export dependence	Exports divided by total industry output	ADB MRIO2021
	Imp	Import dependence	Imports divided by total industry input	ADB MRIO2021
	Vab	Industry added value	Value-added of each manufacturing industry	ADB MRIO2021
	GVC-pat	Participation in global value chains	The sum of GVC forward participation and backward participation	UIBE GVC
	Ratiowair	Proportion of industrial waste gas treatment cost	Industrial waste gas treatment cost divided by total industry treatment cost	CESY

Note: 1. Digitalization input level (Dig): Based on the practice [55,56], in this paper C14 (electrical and optical equipment) and C27 (post and telecommunications) industries in the ADB MRIO2021 input-output table are used as the basic sectors of the digitalization. The ratio of the added value of the two divided by the total added value of the manufacturing industry is used to measure the level of digital input in the manufacturing industry. 2. CEADS: Carbon Emission Accounts & Datasets; IFIND: Financial data terminal of Tonghuashun; CISY: China Industry Statistical Yearbook; CSTSY: China Science and Technology Statistical Yearbook; CESY: China Environmental Statistical Yearbook. 3. The carbon emission data in this paper is the total carbon emission of various energy-related manufacturing industries. The data of electricity consumption is directly derived from the total electricity consumption of various industries in the IFIND database.

**Table 2 ijerph-20-03938-t002:** Manufacturing Industry Segmentation.

Code in This Paper	ADB MRIO Code	Industry	Industry Category
n1	c3	Food, beverages, and tobacco	LI
n2	c4	Textiles and textile products	LI
n3	c5	Leather, leather products, and footwear	LI
n4	c6	Wood and products of wood and cork	LI
n5	c7	Pulp, paper, paper products, printing, and publishing	LI
n6	c8	Coke, refined petroleum, and nuclear fuel	CI
n7	c9	Chemicals and chemical products	TI
n8	c10	Rubber and plastics	LI
n9	c11	Other nonmetallic minerals	CI
n10	c12	Basic metals and fabricated metal	CI
n11	c13	Machinery, nec	CI
n12	c14	Electrical and optical equipment	TI
n13	c15	Transport equipment	TI

Note: LI = labor intensive, CI = capital intensive, TI = technology intensive.

**Table 3 ijerph-20-03938-t003:** Descriptive statistics of variables.

Variable	Obs	Mean	Std. Dev.	Min	Max	Unit
Carbon	169	265.4977	512.6949	0.7527	1942.2960	Mt
Dig	169	0.2275	0.5115	0.0071	2.3739	
Er	169	6.8330	9.9424	0.3462	40.2882	%
Ind	169	19.3438	14.1929	2.8405	78.5029	RMB 10,000/person
Scale	169	0.0500	0.0406	0.0024	0.1730	
Exp	169	0.1188	0.0978	0.0231	0.4605	
Imp	169	0.0632	0.0378	0.0187	0.1985	
Vab	169	1,306,574	867,336	132,607	3,704,156	Million-yuan
GVC-pat	169	0.3109	0.1046	0.1031	0.6068	
Ratiowair	169	4.2115	7.2584	0.0411	30.1847	%

**Table 4 ijerph-20-03938-t004:** Estimation results.

Variable	Core Explanatory Variable	
	(1)	(2)
	Dig-1	Dig-2
Dig	0.453 ***	0.652 **
	(3.35)	(2.87)
Dig^2^	−0.331 **	−0.596 **
	(−2.81)	(−2.66)
Er	0.435 **	0.319 *
	(3.19)	(2.52)
Ind	−0.134 **	−0.196 ***
	(−3.23)	(−4.54)
Scale	−0.278 ***	−0.216 ***
	(−5.64)	(−4.67)
Exp	−0.139 ***	−0.122 **
	(−3.50)	(−3.15)
Imp	0.273 ***	0.328 ***
	(3.82)	(4.25)
Vab	0.0383	0.0840 *
	(0.74)	(2.19)
GVC−pat	−0.354 ***	−0.316 ***
	(−5.02)	(−4.81)
Ratiowair	0.725 ***	0.748 ***
	(6.08)	(6.40)
Obs	169	169
R^2^	0.8954	0.8941

Note: * *p* < 0.05, ** *p* < 0.01, *** *p* < 0.001; *t* statistics in parentheses.

**Table 5 ijerph-20-03938-t005:** Heterogeneity analysis results.

Variable	Capital Intensive		Labor Intensive		Technology Intensive	
	Core Explanatory Variable: Dig					
	(1)	(2)	(3)	(4)	(5)	(6)
	Dig-1	Dig-2	Dig-1	Dig-2	Dig-1	Dig-2
Dig	−2.251 **	−0.049	−1.136	−0.080 ***	−0.081	−0.098
	(−3.23)	(−0.10)	(−1.20)	(−4.92)	(−1.36)	(−0.45)
Dig^2^	3.261	0.117	3.299	0.077 ***	0.047	0.050
	(1.97)	(0.24)	(1.03)	(4.57)	(1.13)	(0.31)
Er	1.302 ***	1.543 ***	0.217 ***	0.220 ***	0.422 ***	0.412 ***
	(5.50)	(5.84)	(6.85)	(8.05)	(6.70)	(7.29)
Controls	Yes	Yes	Yes	Yes	Yes	Yes
Obs	52	52	78	78	39	39
R^2^	0.9720	0.9613	0.9182	0.9357	0.9818	0.9823

Note: ** *p* < 0.01, *** *p* < 0.001; *t* statistics in parentheses.

**Table 6 ijerph-20-03938-t006:** Estimation results of IV-2SLS model.

Variable	Dependent Variable: Carbon	
	Core Explanatory Variable	
	(1)	(2)
	Dig-1	Dig-2
Dig	0.526 ***	0.674 **
	(3.64)	(3.05)
Dig^2^	−0.388 **	−0.617 **
	(−3.13)	(−2.84)
Er	0.453 ***	0.319 **
	(3.41)	(2.61)
Ind	−0.131 **	−0.197 ***
	(−3.26)	(−4.70)
Scale	−0.288 ***	−0.216 ***
	(−5.95)	(−4.82)
Exp	−0.146 ***	−0.123 **
	(−3.76)	(−3.27)
Imp	0.276 ***	0.329 ***
	(3.99)	(4.41)
Vab	0.0311	0.0837 *
	(0.62)	(2.25)
GVC-pat	−0.365 ***	−0.317 ***
	(−5.30)	(−4.99)
Ratiowair	0.719 ***	0.748 ***
	(6.22)	(6.61)
Obs	156	156
R^2^	0.8952	0.8941

Note: * *p* < 0.05, ** *p* < 0.01, *** *p* < 0.001; *t* statistics in parentheses.

**Table 7 ijerph-20-03938-t007:** Threshold effect test results.

Industry	Threshold	Energy Threshold		Economic Threshold		Scale Threshold	
		Threshold Value	*p* Value	Threshold Value	*p* Value	Threshold Value	*p* Value
Labor intensive	Single	−0.5478	0.0100	−0.4233	0.0533	−0.8088	0.0300
	Double	0.0042	0.0300	0.2929	0.2867	−0.9171	0.2000
Capital intensive	Single	0.3053	0.3833	−0.1759	0.6300	−0.5352	0.0000
	Double	−0.4059	0.3967	−0.7593	0.9467	2.2503	0.1900
Technology intensive	Single	−0.2248	0.0000	−0.2786	0.0000	0.1690	0.0000
	Double	0.8068	0.0000	−0.4057	0.6233	1.3146	0.1167

Note: The dependent variable is Carbon.

**Table 8 ijerph-20-03938-t008:** Estimation results of the threshold model.

	Dependent Variable: Carbon						
Industry	Labor Intensive			Capital Intensive	Technology Intensive		
	Energy Threshold	Economic Threshold	Scale Threshold	Scale Threshold	Energy Threshold	Economic Threshold	Scale Threshold
Dig	0.5628	0.7665	−0.7336	4.8318 ***	−0.0696	0.5713 ***	0.2219 **
[<qx1]	(0.8022)	(0.7134)	(0.8482)	(1.6403)	(0.1150)	(0.1283)	(0.0892)
Dig	0.4316	0.5810	−0.6251	2.7802 *	−0.1030	0.0543	0.1301 ***
[>qx1]	(0.8009)	(0.7124)	(0.8383)	(1.5653)	(0.1443)	(0.0678)	(0.0463)
Controls	Yes	Yes	Yes	Yes	Yes	Yes	Yes
Obs	78	78	78	52	39	39	39
R^2^	0.2206	0.2206	0.5048	0.5974	0.4361	0.9490	0.5852

Note: *** significant at 1% level, ** significant at 5% level, * significant at 10% level. qx1 represents the threshold value (standard errors in parentheses).

## Data Availability

The data involved in this study are all from public data.

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
