# Peer review of "Digitalization, Electricity Consumption and Carbon Emissions—Evidence from Manufacturing Industries in China"

_ijerph, 2023, doi:10.3390/ijerph20053938_

Round 1
Reviewer 1 Report
It is a very interesting topic, indeed. The idea is novel and may be advantageous in the practical world of today. Hence based on the topic, and its novelty and need, I recommend this paper for publication. However, there are some minor issues in the paper which the author is required to resolve. The issues are discussed, one by one, below.
1. The title is fine as it clearly depicts the message of the study. Additionally, at the end of the introduction section (the last paragraph), instead of giving the overview of the remaining sections of the paper, conclude the introduction section well, and do the same for each section (i.e., briefly conclude each of them).
2. Moreover, the literature review is smartly written, and it adequately separates different ideas with proper headings, however, all of these ideas are needed to be linked, as well, with the use of linking statements (or sentences), for demonstrating a proper flow of the literature.
3. The paper clearly lacks the sense of paragraphing; some paragraphs are too shorts and consist of only few lines. Kindly rearrange your paragraphs and provide arguments about one main theme in each paragraph.
4. The front-end development can be revamped. I feel that a solid theoretical argument is missing. The literature review appears to have an emphasis on the different studies that are related to the topic and the empirical methods/analysis approach.
5. I solid conceptualization of the key constructs and the theoretical relationship among them would be welcomed.
6. A tad more description on the theoretical implications used would also be good
Reviewer 2 Report
Comments:
1. Simplify and refine the “literature review” section, and add the differences between this study and the previous literature, as well as the improvements and contributions made.
2. Please briefly describe the reasons for choosing the period from 2007 to 2019 as the research period in this study? Are there any specific reasons that need to be explained more clearly?
3. It is suggested to add the “discussion” section to explain the specific reasons for the phenomenon.
4. What are the innovations and limitations of this study?
5. It is suggested to adjust the format of the table (center processing, such as tables 4, 5) to make the essay symmetrical.
6. It is suggested to improve the picture quality and adjust the font of the picture content according to the requirements of the journal.
7. The format and font of the formula part in the text should be standardized.
Reviewer 3 Report
very well written manuscript, however, i suggest few minor revisions.
I suggest to add few more studies in literature review.
The choice of variables should be elaborated.
It is suggested that authors should compare and contrast relevant studies in the discussion section.
Policy recommendations must be expanded.
